

# Handling stress may confound murine gut microbiota studies

Cary R. Allen-Blevins[1], Xiaomeng You[2], Katie Hinde[3,4] and David A. Sela[2,5,6]

[1] Department of Human Evolutionary Biology, Harvard University, Cambridge, MA, United States
[2] Department of Food Science, University of Massachusetts, Amherst, MA, United States
[3] Center for Evolution and Medicine, Arizona State University, Tempe, AZ, United States
[4] School of Human Evolution and Social Change, Arizona State University, Tempe, AZ, United States
[5] Department of Microbiology, University of Massachusetts, Amherst, MA, United States
[6] Center for Microbiome Research, University of Massachusetts Medical School, Worcester, MA, United States

Corresponding author
David A. Sela, davidsela@umass.edu

## ABSTRACT

**Background**. Accumulating evidence indicates interactions between human milk composition, particularly sugars (human milk oligosaccharides or HMO), the gut microbiota of human infants, and behavioral effects. Some HMO secreted in human milk are unable to be endogenously digested by the human infant but are able to be metabolized by certain species of gut microbiota, including *Bifidobacterium longum* subsp. *infantis (B. infantis)*, a species sensitive to host stress (Bailey & Coe, 2004). Exposure to gut bacteria like *B. infantis* during critical neurodevelopment windows in early life appears to have behavioral consequences; however, environmental, physical, and social stress during this period can also have behavioral and microbial consequences. While rodent models are a useful method for determining causal relationships between HMO, gut microbiota, and behavior, murine studies of gut microbiota usually employ oral gavage, a technique stressful to the mouse. Our aim was to develop a less-invasive technique for HMO administration to remove the potential confound of gavage stress. Under the hypothesis that stress affects gut microbiota, particularly *B. infantis*, we predicted the pups receiving a prebiotic solution in a less-invasive manner would have the highest amount of *Bifidobacteria* in their gut.

**Methods**. This study was designed to test two methods, active and passive, of solution administration to mice and the effects on their gut microbiome. Neonatal C57BL/6J mice housed in a specific-pathogen free facility received increasing doses of fructooligosaccharide (FOS) solution or deionized, distilled water. Gastrointestinal (GI) tracts were collected from five dams, six sires, and 41 pups over four time points. Seven fecal pellets from unhandled pups and two pellets from unhandled dams were also collected. Qualitative real-time polymerase chain reaction (qRT-PCR) was used to quantify and compare the amount of *Bifidobacterium*, *Bacteroides*, Bacteroidetes, and Firmicutes.

**Results**. Our results demonstrate a significant difference between the amount of Firmicutes in pups receiving water passively and those receiving FOS actively ($p$-value $= 0.009$). Additionally, we found significant differences between the fecal microbiota from handled and non-handled mouse pups.

**Discussion**. From our results, we conclude even handling pups for experimental purposes, without gavage, may induce enough stress to alter the murine gut microbiota profile. We suggest further studies to examine potential stress effects on gut microbiota

caused by experimental techniques. Stress from experimental techniques may need to be accounted for in future gut microbiota studies.

## INTRODUCTION

The gut microbiota has major physiological and potentially biopsychological implications for human health (*Lyte, 2010*; *Walter & Ley, 2011*; *Oh et al., 2010*; *Allen-Blevins, Sela & Hinde, 2015*). The collection of hundreds of bacterial species in the human gut exerts strong influences on immune function, nutrition, and neurodevelopment through maintaining gut barrier function, fermenting dietary fiber to short-chain fatty acids, and producing neurotransmitters (*Dinan et al., 2015*; *Grenham et al., 2011*). Processes such as programming the immune system likely begin with microbial exposure at birth and perturbations in the gut microbiota early in development have been implicated in chronic disease, including psychological conditions (*Bäckhed et al., 2015*; *Douglas-Escobar, Elliott & Neu, 2013*; *Rook, Lowry & Raison, 2013*).

Recent research in mice suggests early gut microbiota affects neurodevelopment and behavior (*Borre et al., 2014*; *Sudo et al., 2004*; *Diaz Heijtz et al., 2011*). Mice reared without microbiota ("germ-free") exhibit increased corticosterone response to restraint stress and reduced expression levels of brain-derived neurotrophic factor in the hippocampus (*Sudo et al., 2004*). The increased corticosterone response is partially reversed with exposure to *Bifidobacterium longum* subsp. *infantis*, a species dominating the human infant gut (*Sudo et al., 2004*, *Yatsunenko et al., 2012*). Notably, colonization with *B. infantis* only partly normalizes the corticosterone response in 6 week old mice, but not 14 week old mice (*Sudo et al., 2004*). Slightly contrary to this research, germ-free mice showed reduced anxiety behavior in light-dark and elevated maze plus tests (*Diaz Heijtz et al., 2011*). However, the differences in the type of stressor (restraint stress vs. an open field) may cause these contradictory stress responses in germ-free animals. Regardless, Diaz Heijtz and colleagues also demonstrated only early life colonization of germ-free mice, not colonization in mature mice, could normalize the behavior of germ-free mice (*Diaz Heijtz et al., 2011*). These studies suggest critical neurodevelopmental windows exist in early life during which gut microbiota are crucial to shaping behavior (*Borre et al., 2014*; *Allen-Blevins, Sela & Hinde, 2015*).

### Evolutionary context

If early life gut microbiota are critical for normal neurodevelopment, mothers necessarily play an essential role in programming neurodevelopment through transmitting and supporting the microbiota (*Bäckhed et al., 2015*; *Allen-Blevins, Sela & Hinde, 2015*; *Sela & Mills, 2010*). Initial microbial colonization is vertically transmitted from mother to offspring during delivery (*Bäckhed et al., 2015*; *Dominguez-Bello et al., 2010*;

*Mueller et al., 2014; Hinde & Lewis , 2015*). The newly colonized infant gut is then exposed to mother's milk, which in humans contains glycans such as human milk oligosaccharides (HMO) that are not digested by the infant (*Marcobal & Sonnenburg, 2012; Sela & Mills, 2010*). While the infant does not possess endogenous enzymes to cleave HMO, certain species of gut microbiota can metabolize HMO, including *B. infantis* (*Sela & Mills, 2010; Sela et al., 2008; Sela et al., 2011; Sela et al., 2012*). *B. infantis* is capable of metabolizing HMO as a sole carbon source, and *B. infantis, Bifidobacterium longum* and *Bifidobacterium breve* affect stress and anxiety behaviors (*Yatsunenko et al., 2012; Sudo et al., 2004; Savignac et al., 2014; Sela & Mills, 2010; Sela et al., 2008; Desbonnet et al., 2010*). Particular strains of *Bifidobacterium* can also produce $\gamma$-aminobutyric acid (GABA), a major inhibitory neurotransmitter (*Barrett et al., 2012; Yunes et al., in press*). The resulting interactions create a milk-microbiota-brain-behavior (M2B2) system, which may allow mothers to influence infant behavior through their milk (*Allen-Blevins, Sela & Hinde, 2015*).

## Experimental rationale

Studying the M2B2 system in model organisms presents unique challenges because experimental techniques can induce stress in animals that affects microbiota and behavior (*Hoggatt et al., 2010; Bailey & Coe, 1999; Bailey, Lubach & Coe, 2004*). Stress is a challenge to homeostasis which may be caused by environmental, physiological, social, or psychological stimuli (*Bailey, 2014, Mendoza, in press*). Early life stress, including neonatal handling and maternal separation, can have long-term developmental consequences and disrupt the regulation of crucial biopsychological pathways, such as the hypothalamus-pituitary-adrenal axis (*Dalmaz et al., 2015; O'Mahony et al., 2009*). Gut microbiome experiments frequently involve oral gavage of rodents with known bacterial strains, fecal matter, or other compounds (*Turnbaugh et al., 2006; Fujimura et al., 2014; Ji et al., 2012*). This technique induces stress responses and can be injurious or fatal to mice, particularly when they are very young (*Hoggatt et al., 2010; Flamm, 2012*). Alterations in gut microbiota in response to host stress have been demonstrated in mice and rhesus macaques (*Tarr et al., 2015; Bailey, Lubach & Coe, 2004; Bailey & Coe, 1999*). If the gut microbiota is sensitive to stress, invasive techniques, like gavage, introduce a potential confound. Changes in microbial profiles over the course of an experiment could be due to the treatment or stress induced from experimental techniques. Since the M2B2 system must be studied prior to weaning, methods such as dosing water or chow with HMO are not effective. Therefore, a non-invasive technique for administering prebiotic solutions directly to very young mice is necessary.

The purpose of this experiment was to determine a less stressful method for administering experimental prebiotic liquids to conventional mouse pups. Our aims were to develop a method of studying particular diet-microbe interactions in non-gnotobiotic mice. Reducing psychological perturbation was a main goal because bifidobacteria that dominate the infant gut microbiome are reduced after stress exposure (*De Leoz et al., 2014; Bailey, Lubach & Coe, 2004*). Therefore, we predicted pups receiving a prebiotic solution in a more passive manner would have higher amounts of *Bifidobacteria.* We tested two methods, active and passive, of administering fructooligosaccharide (FOS), a previously demonstrated

bifidogenic prebiotic (*Howard et al., 1995*), to mouse pups from post-natal day 1 to post-natal day 21 (PND1-PND21). FOS was used in this pilot experiment due to its bifidogenic properties and the prohibitive cost of HMO. *Bifidobacterium* and *Bacteroides* counts were analyzed because of their potential roles in neurodevelopment (*Allen-Blevins, Sela & Hinde, 2015*; *Hsiao et al., 2013*; *O'Sullivan et al., 2011*), while Bacteroidetes and Firmicutes were analyzed due to these phyla being dominant within human gut microbiomes.

## MATERIALS & METHODS

### Subjects

We conducted our methodological study (Fig. 1) in captive-bred laboratory mice (*Mus musculus*). Six timed-pregnant C57BL/6J females and six C57BL/6J males were purchased from The Jackson Laboratory (Bar Harbor, ME, USA) at six weeks old. Animals were housed in the Harvard University Biological Research Infrastructure, a specific-pathogen free facility, under standard Institutional Animal Care and Usage Committee (IACUC) murine environmental conditions. Water and PicoLab commercial chow were available *ad libitum*. In consideration of the greater risk of reduced maternal care and increased pup mortality among C57BL/6J primiparae (*Brown et al., 1999*), initial litters were culled and dams placed in single pair mating cages for one week. These mating pairs produced the litters used for the experimental manipulations. One dam may have still been nulliparous, as she exhibited signs of pregnancy but no litter was observed prior to being placed in a mating cage. However, the litter may have been delivered and cannibalized prior to observation. This would be consistent with other first litters from these dams being cannibalized or found dead. Due to the unexpected death of one male, M1 was mated to F1 and then to F6. The other matings were as follows: M2–F2, M3–F4, M4–F3, and M5–F5. Animals were maintained in breeding cages for seven days before females were removed to individual cages for experimental manipulations. Males remained in single, separate home cages, undisturbed except for cage changes, until euthanasia approximately one week after the birth of their sired litter. All cages were clear plastic, 10.5 inches by 6.5 inches by 5.0 inches.

### Experimental manipulations

Dams were randomly assigned to the following experimental groups: passive water, passive fructooligosaccharide (FOS), active water, active FOS, buccal water, and buccal FOS. The litters for each group were reduced to six on PND0, with the exception of the active water litter, which only contained five pups at birth. The pregnancy of the buccal water dam failed, leaving no experimental litter for the condition.

Pups in each experimental group were handled daily. At the beginning of each provision of the assigned treatment, the home cage of the litter was placed next to a clean cage with fresh bedding. The dam was removed from the home cage and placed into the clean cage for the duration of pup manipulation for that day. Pups were immobilized by grasping the skin at the nape and along the spine, and rotating their bodies to reveal the ventrum. Starting at PND10, pups were lifted by the tails prior to grasping the nape. For active FOS and water conditions, a micropipette tip was placed in the pup's mouth and the dosage was injected directly into the oral cavity. In passive conditions, a micropipette tip was used to

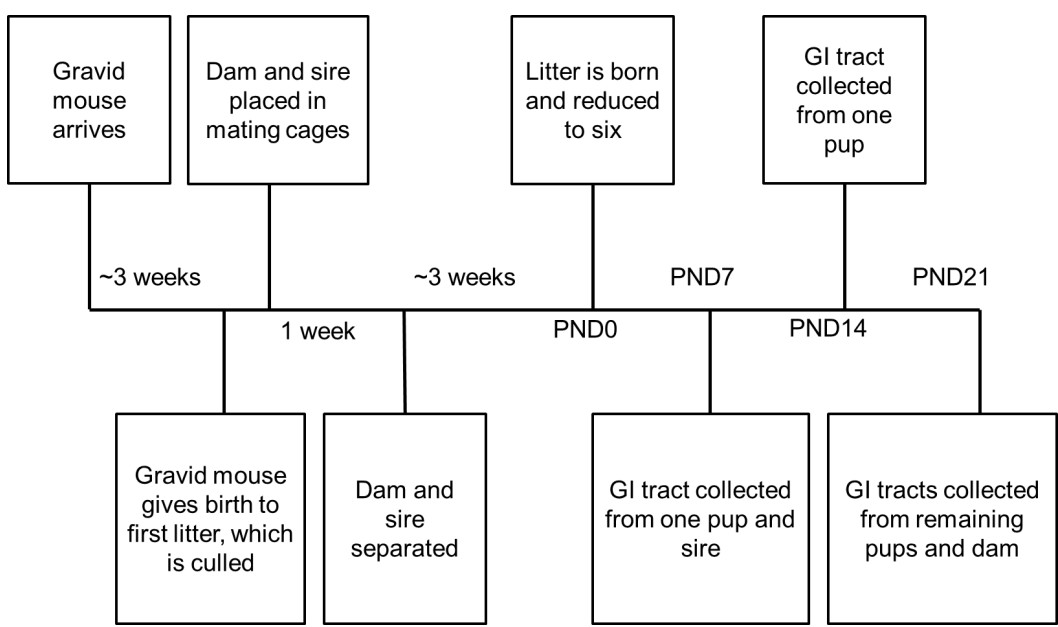

**Figure 1  Timeline for each treatment group.** Samples were collected at time points post-natal day (PND)0, PND7, PND14, and PND21.

transfer the dosage to a Crematocrit tube that was then placed near the pup's mouth with the intent to induce the suckling response (*Szczypka et al., 1999*). As pups in the passive litters aged and the dosages increased (Fig. 2), only micropipette tips were used to place the dosage near the pups' mouths. The switch to only micropipette tips occurred on PND12 for the passive water group and PND15 for the passive FOS group. All tips and tubes were autoclaved prior to use. For buccal conditions, the daily dosage was micropipetted onto a sterile cotton swab, which was then inserted into the pup's mouth. After receiving the daily dosage, the pup was returned directly to the home cage and the next pup was removed for dosing. Once pups began to open their eyes, they were placed into the clean cage with the dam after dosing. When all pups had received their daily treatment, the dam and pups were returned to the home cage.

FOS was purchased from Sigma-Aldrich Corp. and administered in a 2.5mM solution for PND1-PND7and a 25mM solution for PND8-PND21. 25mM was the concentration of HMO producing results when given to mice from birth to weaning in *Kurakevich et al. (2013)*. The 2.5mM dosage was used to determine whether the 25mM could be further reduced, to decrease future HMO cost. Distilled, deionized water was used to create the FOS solutions. Distilled, deionized water was also used for the water conditions.

## Sample collection

Gastrointestinal tracts (GI) were collected from experimental groups on PND0, PND7, PND14, and PND21, while control fecal samples from non-handled, non-dosed mice were collected on PND14 and PND20. On PND0 as many pups as necessary to reduce litter size to six were anesthetized with carbon dioxide, decapitated with sharp scissors, and their GI tracts were collected. Since the active water litter had five pups at birth, only one

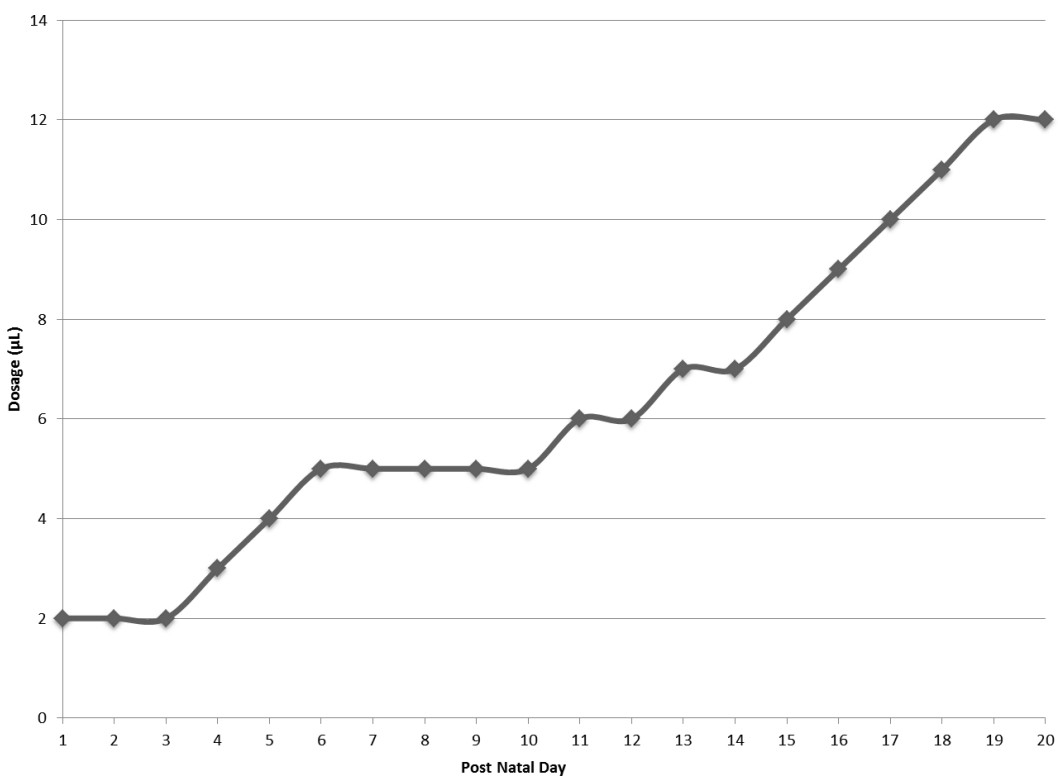

**Figure 2 Daily dosage of FOS or water in microliters.** Dosage of water or FOS increased from 2 microliters ($\mu$L) to 12 microliters ($\mu$L) over the course of the experiment.

pup was euthanized on PND0. On PND7, one pup from each litter was euthanized in the same manner and their GI tract collected. At PND14, one pup from each litter was euthanized with carbon dioxide and their GI tract collected. On PND21, all remaining pups in the litter were euthanized with carbon dioxide and their GI tracts collected. Dams were also euthanized with carbon dioxide on PND21 and GI tracts were collected. Sires were euthanized with carbon dioxide and GI tracts were collected approximately one week after the birth of their litter. GI tracts were snap-frozen in dry ice and stored at $-80\,°C$ until analysis. Voided fecal pellet samples were also collected on PND14 and PND20 from non-handled C57BL/6J pups and two non-handled dams housed in the same facility, matched for living conditions and diet, which served as controls to the treatment groups. Animal use was approved by the Harvard University Institutional Animal Care and Use Committee under protocol 14-08-217.

## qRT-PCR analysis

Fecal pellet and GI tract material was transferred from Harvard University, Cambridge, Massachusetts on dry ice to University of Massachusetts-Amherst, Amherst, Massachusetts for quantitative real-time polymerase chain reaction (qRT-PCR) analysis. GI tracts were thawed on ice and fecal material from the tracts was scraped into sterile sample tubes. Samples were unattainable from PND0 GI tracts ($N = 9$), due to lack of fecal matter within

the tracts. Additionally, samples from the buccal FOS group (F6, $N = 10$) were not analyzed because there was no litter from F3 (the buccal water dam).

DNA was extracted using a bead beating protocol (FastPrep-24$^{TM}$ 5G MP Biomedicals Inc, US) and standard protocol for the QIAmp DNA stool kit (Qiagen, Valencia, CA, US). DNA quality was determined via nanodrop (Thermo Fisher Scientific, Waltham, MA, US) and extracted DNA was diluted to 4 ng/µL. Custom TaqMan gene expression assays (Thermo Fisher Scientific) for *Bifidobacterium*, Firmicutes, *Bacteroides*, and Bacteroidetes were designed using the following sequences:

> *Bifidobacterium* (*Penders et al., 2005*):
> Forward primer: GCGTGCTTAACACATGCAAGTC
> Reverse primer: CACCCGTTTCCAGGAGCTATT
> Probe: TCACGCATTACTCACCCGTTCGCC
> Firmicutes (*Lecerf et al., 2012*):
> Forward primer: GAATCTTCCACAATGGAC-GAAAG
> Reverse primer: AATACCGTCAATACCTGAACAGT-TACTC
> Probe: CTGATGGAGCAACGCCGCGT
> *Bacteroides* (*Layton et al., 2006*):
> Forward primer: GAGAGGAAGGTCCCCCAC
> Reverse primer: CGCTACTTGGCTGGTTCAG
> Probe: CCATTGACCAATATTCCTCACTGCTGCCT
> Bacteroidetes (*Dick & Field, 2004*):
> Forward primer: AACGCTAGCTACAGGCTTAACA
> Reverse primer: ACGCTACTTGGCTGGTTCA
> Probe: CAATATTCCTCACTGCTGCCTCCCGTA

Samples were run in triplicate, with negative control blanks of RNAse free water and a standard curve included on each 96-well plate, using Applied Biosystems 7500 Fast Real-Time PCR system (Thermo Fisher Scientific). Wells included 1 µL of TaqMan gene expression assay, 10 µL of TaqMan master mix (Thermo Fisher Scientific), 5 µL of RNAse free water, and 4 µL of the DNA sample for a total of 20 µL in each well. Plates were run at 50C for 2 min, 95C for 10 min, and then 45 cycles of 95C for 15 s and 60C for 1 min.

## Statistical analysis

The Kruskal-Wallis test was performed in RStudio (Version 0.98.1103) to compare variation in *Bifidobacterium*, Firmicutes, *Bacteroides*, and Bacteroidetes quantity between treatment groups: active fructooligosaccharide (FOS), passive FOS, active water, and passive water. Wilcoxon rank sum tests were performed in RStudio (Version 0.98.1103) to compare variation in *Bifidobacterium*, *Bacteroides*, and Bacteroidetes quantity in the combined totality of the treatment groups (active FOS, passive FOS, active water, and passive water) versus the fecal pellet samples from non-handled mice housed in the same facility, for both pup and dam samples. Since the sample sizes were very unequal for non-handled control ($N = 7$) versus treatment ($N = 21$) pups, Wilcoxon tests were also performed to compare the quantity of *Bifidobacterium*, *Bacteroides*, and Bacteroidetes in the combined

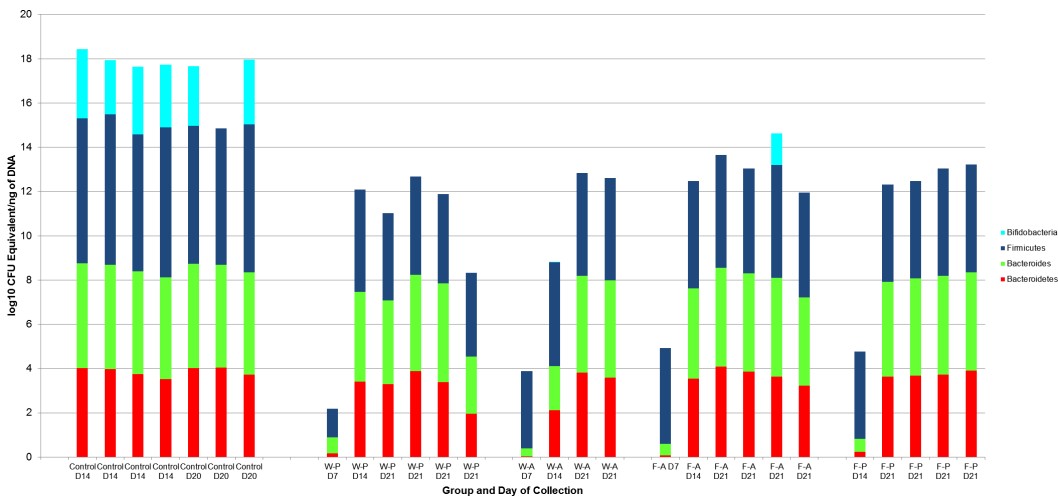

**Figure 3** **log10 Colony-forming unit (CFU) equivalents/ng of sample DNA for control and treatment groups.** Wilcoxon rank sum test demonstrated significant differences between the control and treatment samples for all *Bifidobacterium* ($p < 0.001$), *Bacteroides* ($p < 0.001$), and Bacteroidetes ($p$-value $= 0.008$). There was also a significant difference for Firmicutes between the WP and FA groups ($p$-value $= 0.009$). W-P, water passive, W-A, water active; F-A, FOS active; F-P, FOS passive.

treatment groups against the fecal pellet samples from pups on PND14 and PND21. For PND14, sample sizes were equal at four control and four treatment pups. As no fecal pellet samples were collected on PND21 and no GI tracts were collected on PND20, the fecal pellet samples from PND20 were compared to GI tract samples from PND21. Significance for the Kruskal-Wallis and Wilcoxon tests was established as $p < 0.05$. One sample, the FOS passive PND7, was excluded from analysis due to bacterial detection levels below the standard curve.

## RESULTS

### Gastrointestinal (GI) tract samples

A log10 colony-forming unit (CFU) equivalents/ng of DNA were measured for *Bifidobacterium*, *Bacteroides*, Bacteroidetes, and Firmicutes for each pup treatment group ($N = 21$, Fig. 3). All treatment group samples demonstrated some amount of Bacteroidetes, *Bacteroides*, and Firmicutes. Maximum, minimum, and median counts, as well as the interquartile range, for each taxa are listed in Table 1. One data point was excluded from the *Bifidobacterium* counts due to a lack of agreement among triplicate samples (active water, PND7). From a total of 20 treatment samples, qRT-PCR revealed the majority (19) to have $< 10^1$ quantity of bifidobacteria. The Kruskal-Wallis test determined no significant difference in the median bacterial counts across treatment groups for Bacteroidetes ($p = 0.546$), *Bacteroides* ($p = 0.534$), or *Bifidobacterium* ($p = 0.786$). However, there was a significant difference for Firmicutes ($p = 0.043$).

To determine which treatment groups were significantly different in the amount of Firmicutes, Wilcoxon rank sum tests were performed in a pairwise fashion. There were no significant differences in Firmicutes amounts between the passive water and active

**Table 1  log 10 Colony-forming unit (CFU) equivalents/ng of sample DNA for pup samples.** Maximum, minimum, median, and interquartile range (IQR)counts for Bacteroidetes, *Bacteroides*, *Bifidobacterium*, and Firmicutes in treatment (GI tract, $N = 21$) and control (Fecal pellets, $N = 7$) pups.

| Group/Taxa | Maximum | Minimum | Median | Interquartile range |
|---|---|---|---|---|
| GI Tract—Pups ($N = 21$) | | | | |
| Bacteroidetes | $10^4$ | $<10^1$ | $10^3$ | $10^1$ |
| *Bacteroides* | $10^4$ | $<10^1$ | $10^4$ | $10^1$ |
| *Bifidobacterium* | $10^1$ | $<10^1$ | $<10^1$ | $<10^1$ |
| Firmicutes | $10^5$ | $10^1$ | $10^4$ | $<10^1$ |
| Fecal Pellets—Pups ($N = 7$) | | | | |
| Bacteroidetes | $10^4$ | $10^3$ | $10^3$ | $<10^1$ |
| *Bacteroides* | $10^4$ | $10^4$ | $10^4$ | $<10^1$ |
| *Bifidobacterium* | $10^3$ | $<10^1$ | $10^2$ | $<10^1$ |
| Firmicutes | $10^6$ | $10^6$ | $10^6$ | $<10^1$ |

water groups ($p = 0.171$), passive water and passive FOS groups ($p = 0.247$), active water and active FOS groups ($p = 0.067$), or active FOS and passive FOS groups ($p = 0.329$). However, there was a significant difference between the median Firmicutes counts for the passive water ($10^3$) and active FOS groups ($10^4$; $p = 0.009$).

Bacterial counts were also quantified for samples from the GI tracts of the litter sires ($N = 4$, Fig. 4) and the dams of each treatment group and the buccal water dam with no litter ($N = 5$, Fig. 5). Maximum, minimum and median counts, along with the interquartile range, are in Table 2. The maximum *Bifidobacterium* count for the sires came from the sire mated to the buccal water dam, which produced no litter. All samples for the treatment dams contained $<10^1$ *Bifidobacterium*.

## Fecal pellet samples

Fecal pellet samples from non-handled pups and dams were measured for *Bifidobacterium*, *Bacteroides*, Bacteroidetes, and Firmicutes log10 CFU equivalents/ng of DNA as a control (Fig. 3 for pups, Fig. 5 for dams). Maximum, minimum, and median counts, with the interquartile ranges, can be found in Table 1 for the pups, while the same counts for the dams can be found in Table 2. The maximum *Bifidobacterium* count of $10^3$ was found in two pup samples. For the dams, both samples contained $10^1$ *Bifidobacterium*.

## Gastrointestinal tract (GI) versus fecal pellet samples

Since there were no significant differences in bacterial counts of Bacteroidetes, *Bacteroides*, or *Bifidobacterium* across treatment groups, we combined these groups for analysis against the fecal pellet controls. The Wilcoxon rank sum test revealed significant differences between counts for all bacterial taxa (*Bifidobacterium p* value $< 0.001$, Bacteroidetes *p* value $= 0.008$, *Bacteroides p* value $< 0.001$) in the treatment groups and bacterial counts for the fecal pellets collected from pups (Fig. 3).

Comparisons by post-natal day also revealed significant differences between non-handled and handled pups. Wilcoxon rank sum tests demonstrated significant differences in the medians of the control and treatment samples for *Bacteroides* (control median: $10^4$

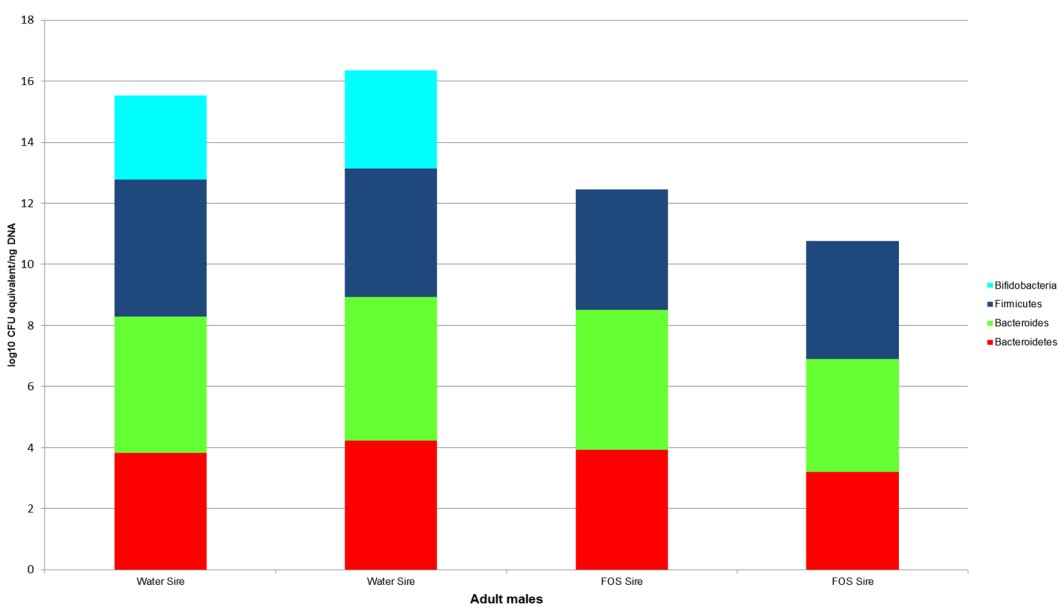

**Figure 4 log10 colony-forming unit (CFU) equivalents/ng of sample DNA for feces collected from the GI tracts of treatment litter sires.** Samples from water sires contained *Bifidobacteria*, while samples from FOS sires did not.

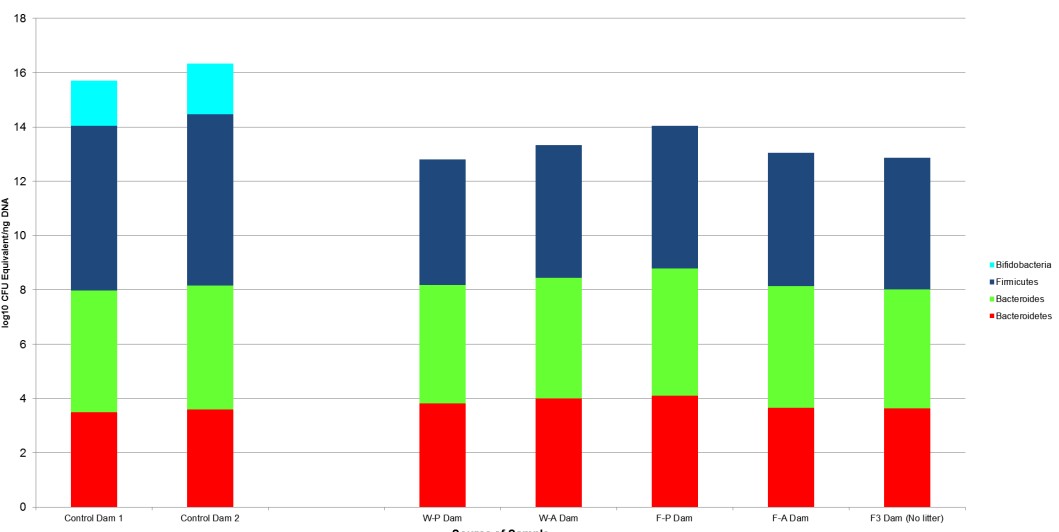

**Figure 5 The log10 colony-forming units (CFU) equivalents/ng of sample DNA for control and experimental dams.** There was no significant difference between the samples, despite the treatment dams having no bifidobacteria. The F3 dam was the buccal water dam that did not deliver a litter. W-P, passive water; W-A, active water; F-A, active FOS; F-P, passive FOS.

(IQR $= <10^1$), treatment median: $10^3$ (IQR $= 10^2$), $p$-value $= 0.03$) and *Bifidobacterium* (control median: $10^2$ (IQR $= <10^1$), treatment median: $< 10^1$ (IQR $=< 10^1$), $p$-value $= 0.03$) on PND14, but a non-significant difference for Bacteroidetes ($p$-value $= 0.057$, Fig. 6). These significant differences for *Bacteroides* ($p$-value $=0.003$) and *Bifidobacterium*

**Table 2  The log 10 colony-forming unit (CFU) equivalents/ng of sample DNA for dams and sires.** Maximum, minimum, median and interquartile range (IQR) counts for Bacteroidetes, *Bacteroides*, *Bifidobacterium*, and Firmicutes in sires, and treatment and control dams.

| Group/Taxa | Maximum | Minimum | Median | Interquartile range |
|---|---|---|---|---|
| GI Tract—Dams ($N = 5$) | | | | |
| Bacteroidetes | $10^4$ | $10^3$ | $10^3$ | $<10^1$ |
| *Bacteroides* | $10^4$ | $10^4$ | $10^4$ | $<10^1$ |
| *Bifidobacterium* | $<10^1$ | $<10^1$ | $<10^1$ | $<10^1$ |
| Firmicutes | $10^5$ | $10^4$ | $10^4$ | $<10^1$ |
| GI Tract—Sires ($N = 4$) | | | | |
| Bacteroidetes | $10^4$ | $10^3$ | $10^3$ | $<10^1$ |
| *Bacteroides* | $10^4$ | $10^3$ | $10^4$ | $<10^1$ |
| *Bifidobacterium* | $10^3$ | $<10^1$ | $10^1$ | $10^2$ |
| Firmicutes | $10^4$ | $10^3$ | $10^4$ | $<10^1$ |
| Fecal Pellets—Dams ($N = 2$) | | | | |
| Bacteroidetes | $10^3$ | $10^3$ | $10^3$ | $<10^1$ |
| *Bacteroides* | $10^4$ | $10^4$ | $10^4$ | $<10^1$ |
| *Bifidobacterium* | $10^1$ | $10^1$ | $10^1$ | $<10^1$ |
| Firmicutes | $10^6$ | $10^6$ | $10^6$ | $<10^1$ |

($p$-value $= 0.006$) median counts in the control and treatment samples were also evident at PND21, despite a large difference in sample size (control $N = 3$, treatment $N = 14$).

Wilcoxon rank sum tests found no significant difference between samples from the two control dams and samples from the treatment dams ($N = 4$). This lack of significance was maintained when the F3 dam, who had no litter and received no treatment, was included (Fig. 5).

# DISCUSSION

In this experiment, we specifically focused on *Bifidobacterium*, Firmicutes, *Bacteroides*, and Bacteroidetes, phyla frequently studied in animal models due to their presence and hypothesized importance in the human gut. While quantifying and studying the gut microbiota as a whole is a necessary step to fully understand interactions between the host and microbiota, we narrowed our focus to these phyla because of our particular aims for this study. *Bifidobacterium* and *Bacteroides* are both good candidate genera for containing species that potentially affect neurodevelopment during infancy. Both genera have previously been shown to contain species, such as *Bifidobacterium longum* subsp. *infantis* and *Bacteroides fragilis*, that can affect behavior, possibly through neurodevelopmental pathways (*Allen-Blevins, Sela & Hinde, 2015*; *Sudo et al., 2004*; *Hsiao et al., 2013*). Firmicutes and Bacteroidetes were also quantified due to their correlation with obesity, inverse correlation with each other, and frequent measurement in gavage experiments (*Turnbaugh et al., 2006*; *Li et al., 2015*).

The major finding of this paper is the significant decrease in *Bifidobacterium*, *Bacteroides*, and Bacteroidetes present in the gut microbiota of handled animals provided either water

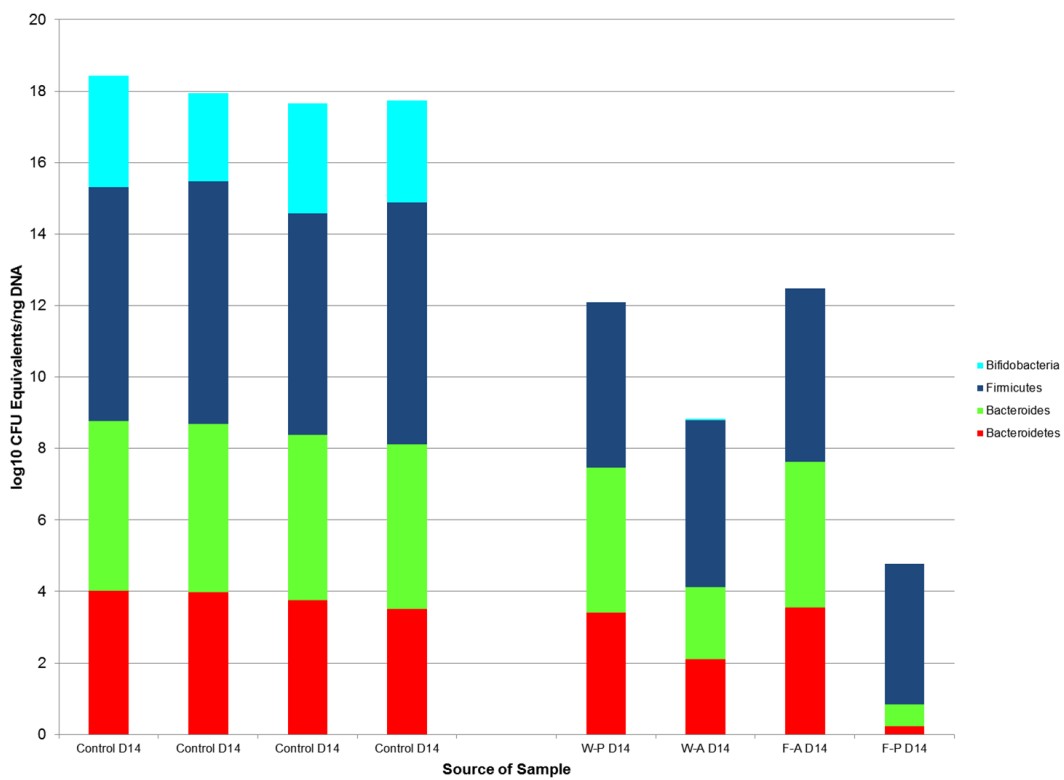

**Figure 6** **The log10 colony-forming unit (CFU) equivalents/ng of sample DNA from control and treatment samples collected on post-natal day 14.** The median *Bifidobacterium* and *Bacteroides* counts in the control samples were significantly different from the treatment samples (*p*-value = 0.03 for both comparisons) as tested by Wilcoxon rank sum test. The medians of Bacteroidetes counts were not significantly different between the treatment and control groups (*p*-value = 0.057). Firmicutes could be compared between control and treatment groups, because there were significant differences between treatment groups for this taxa. W-P, passive water; W-A, active water; F-A, active FOS; F-P, passive FOS.

or FOS. Cognizant of stress effects on bifidobacteria (*Bailey, Lubach & Coe, 2004*), our experiment was designed to determine a less-invasive technique to administer bifidogenic compounds to neonatal mice. While the ultimate goal was to create a method for administration of human milk oligosaccharides (HMO) to mice, FOS was used in this initial experiment due to the prohibitive cost of HMO. We expected to find the most bifidobacteria in the group passively fed FOS, since this was expected to be the least stressful technique and included supplementation of a compound previously demonstrated as bifidogenic (*Howard et al., 1995*). However, there were no significant differences in Bacteroidetes, *Bacteroides*, or *Bifidobacterium* across our treatment groups, falsifying our prediction. There was a significant difference between the median amount of Firmicutes in the pups provided water passively and the pups actively provided with FOS. These two treatment groups were different on both factors (treatment method and substance), so the significant difference is not suprising. Additionally, administration of FOS is correlated with an increase in Firmicutes (*Li et al., 2015*). The median amounts of Firmicutes across treatment groups remained between $10^3$ and $10^4$, while the control pups had a median amount of $10^6$. While we were unable to statistically compare these values due to the difference in treatment

groups for Firmicutes, there seems to be a substantial difference between the amount of Firmicutes in the control and treatment groups. The statistically significant differences in quantities of *Bacteroides*, Bacteroides, and *Bifidobacterium* between the control and treatment groups suggest a factor common to all four treatment groups may have affected these phyla of gut microbiota.

Lack of initial exposure to *Bifidobacterium, Bacteroides*, and Bacteroidetes and differences between voided fecal pellets and GI tract fecal samples may explain the differences in bacterial quantities between our handled and non-handled animals; however, we think these are unlikely explanations. The absence of bifidobacteria in the handled dams creates the possibility of the handled pups lacking bifidobacteria because they had no exposure. However, the counts of *Bifidobacterium* in the water sires indicate at least the dams for the water groups were exposed to bifidobacteria for seven days. Sires were cohoused with the dams for one week and the coprophagic habit of mice (*Heinrichs, 2001*) makes dam exposure to bifidobacteria highly likely. While both the FOS sires and dams appear to be lacking bifidobacteria, one of the FOS pups had the highest count of bifidobacteria in the treatment groups ($>10^1$). Therefore, non-exposure to bifidobacteria is unlikely to explain our results. Non-exposure also cannot explain the differences in *Bacteroides* and Bacteroidetes, as all of the treated dams contained these taxa. Additionally, the differences in the use of voided fecal pellet samples from the control animals (collected from cages) and internal fecal samples from the treatment animals (collected from the distal colon of GI tracts) are also unlikely to explain the magnitude of difference in bifidobacteria. Fecal samples in both mice and humans have demonstrated much higher bifidobacterial counts than gut lumen samples (*Marcotte & Lavoie, 1996*; *Ouwehand et al., 2004*). However, our treatment samples were of intact feces collected from the distal colon and not samples of the mucosa or luminal fluid. Also, the two water sires had counts of *Bifidobacterium* in stool from their intestinal tract that were greater than the counts in the voided fecal pellets of the control dams. Therefore, while some variation may be expected due to differences in sample collection, it is unlikely to reach the magnitude of difference between the median bifidobacteria from the control pups and the treatment pups.

While non-exposure to bacteria is an unlikely explanation, the stress of handling, common to all treatment groups, may have contributed to the changes across the treatment group gut microbiota. *Bifidobacterium*, known to be susceptible to host stress (*Bailey, Lubach & Coe, 2004*), was decreased to the point of $>10^1$ in the majority of our treated mice, the largest decrease of any of the bacterial taxa in our study. Since all of the control animals and two of our minimally handled sires largely maintained *Bifidobacterium* in their gut, and all animals were of the same genetic background, in the same facility on the same diet, the sharp decrease in this taxa, known to be stress-sensitive, is likely due to a stressor. It should be noted that this explanation may not be valid for the lack of *Bifidobacterium* in the FOS sires. Additionally, *Lactobacillus*, a genus within the Firmicutes phylum is also stress-sensitive (*Bailey, 2014*). While the significant difference in Firmicutes between the passive water pups and active FOS pups suggests administration of FOS was able to increase the amount of Firmicutes, all of the treatment groups still had a median amount $10^2$–$10^3$ below the median amount of the control group. Therefore, handling stress

could potentially be driving decreases in the other bacterial taxa as well, to varying degrees. Both passive and active handling techniques appear to negatively influence the amount of *Bifidobacterium*, *Bacteroides*, and Bacteroidetes in the guts of our treated subjects, with the quantity of *Bifidobacterium* being the most severely affected.

If even non-invasive handling stress is potentially correlated with significant changes to the gut microbiota and the loss of an entire taxon, this creates a particular challenge for research centered on the gut microbiota and early life development. To adequately study a potential milk-microbiota-brain-behavior (M2B2) pathway, supplementation of animal models with HMO may be necessary (*Allen-Blevins, Sela & Hinde, 2015*). While gavage is frequently used to administer compounds to rodents, this technique can stress the animal (*Flamm, 2012*; *Hoggatt et al., 2010*). Stress in early life, such as maternal separation, is correlated with significant changes gut microbiota (*Bailey & Coe, 1999*; *O'Mahony et al., 2009*). If non-invasive handling is provoking a stress response in laboratory animals significant enough to affect gut microbiota, then handling young animals during microbiota experiments may confound the results. For example, handling all of the animals, including controls and those receiving vehicles, may lead a researcher to conclude there are none or reduced levels of a bacterium that may actually be diminished due to handling stress. Such a reduction in bacterial taxa may mask potential interactions between an experimental compound and bacterial taxa that might be present if the compound was administered without stress. More research is necessary to determine if common experimental techniques are creating confounds in murine microbial studies.

Synbiotics, a combination of prebiotics and the bacteria of interest (*Schrezenmeir & De Vrese, 2001*), and communal use of non-handled control animals may mitigate the potential challenges of handling stress affecting gut microbiota. Since synbiotics provide both the substrate for bacterial growth and the bacteria (*Schrezenmeir & De Vrese, 2001*), they may provide a method for studying the most stress sensitive bacteria. For both human and animal studies, exposing the subject to the desired bacteria daily may continually replace the diminished strains and mimic the effects of permanent colonization for the duration of exposure. Microbial and other changes can then be compared in these animals to non-handled control animals. As a guiding principle of the American Association for Laboratory Animal Science (*Committee for the Update of the Guide for the Care and Use of Laboratory Animals, 2011*) is understandably to reduce the number of animals used in experiments, there may be a reluctance to include negative control animals that are not handled at all. The reduction principle can still be achieved for gut microbiota studies if multiple labs share feces and potentially other data from non-handled animals housed in the same facilities, on the same diets, and from the same genetic backgrounds.

Our study has several caveats and limitations. It is important to note the small sample sizes in our study. The small number of control samples compared to the treatment samples may have impacted our results. To determine if handling is truly causing such a large difference in gut microbiota, replicating this study with a greater number of control samples would be of great value. We also did not take corticosterone measurements of the mice, which would have allowed us to quantify their physiological stress reaction to handling. Additionally, the stress of handling does not explain the lack of bifidobacteria in

the FOS sires or the buccal water dam. Since this dam had no litter, she was also minimally handled. However, these animals contained counts of Bacteroidetes, *Bacteroides*, and Firmicutes on par with their sex and age-matched conspecifics. After removal from mating cages, all dams were housed separately until parturition. While isolation of pregnant females to prevent cannibalism of pups is an accepted practice (*Committee for the Update of the Guide for the Care and Use of Laboratory Animals, 2011*), since this female never had a litter, the social stress of isolation may have affected her differently. The FOS sires may have lacked bifidobacteria due to an unnoticed illness or a difference in genotype (*Bevins & Salzman, 2011*; *Wacklin et al., 2011*), but this is simply speculation. Therefore, since these two animals present a conundrum and it is highly important to determine the extent of handling effects on gut microbiota, this research should be repeated with larger sample sizes and including measures of corticosterone. Prior to that, researchers should remain cognizant of potential handling effects on their data.

Additionally, we used quantitative real-time PCR (qRT-PCR) to quantify the bacteria number in our samples, which is regarded as a sensitive and specific method to detect commensal bacteria (*Castillo et al., 2006*). Though qRT-PCR is regarded as an accurate method, we found some samples to contain more *Bacteroides* than Bacteroidetes which might be due to qRT-PCR amplification bias. Thus, primers should be carefully designed to ensure the same amplification efficiency among the bacteria of interest in future studies.

## CONCLUSION

Although more research is clearly necessary, the stress of handling, or even the social stress of isolation, may have the capacity to affect murine gut microbiota. Particularly when using young animals to investigate microbial responses to prebiotics, such as studies focusing on the potential milk-microbiota-brain-behavior (M2B2) system, care should be taken during experiments to ensure necessary controls and accurate data collection. Sharing fecal samples from control animals of the same genetic background, housed in the same facility, and fed the same diet would met standards of animal reduction, while enabling comparison of handled treatment animals to non-handled animals.

## ACKNOWLEDGEMENTS

The authors thank Prof. Rachel Carmody for providing the fecal pellet control samples and Harvard University's Institute for Qualitative Social Science for statistical guidance. Our gratitude to the three anonymous reviewers for their comments that improved the manuscript.

### Funding

Funding was partially provided by a grant from Harvard University's Mind, Brain, and Behavior Program. CRAB is supported by a National Science Foundation Graduate Research Fellowship (NSF DFE1144152). XY was partially supported by the Stanley Charm

Graduate Fellowship. The funders had no role in study design, data collection and analysis, decision to publish, or preparation of the manuscript.

### Grant Disclosures
The following grant information was disclosed by the authors:
Harvard University's Mind, Brain, and Behavior Program.
National Science Foundation Graduate Research Fellowship: NSF DFE1144152.
Stanley Charm Graduate Fellowship.

### Competing Interests
The authors declare there are no competing interests.

### Author Contributions
- Cary R. Allen-Blevins conceived and designed the experiments, performed the experiments, analyzed the data, wrote the paper, prepared figures and/or tables, reviewed drafts of the paper.
- Xiaomeng You performed the experiments, analyzed the data, reviewed drafts of the paper.
- Katie Hinde and David A. Sela conceived and designed the experiments, analyzed the data, reviewed drafts of the paper.

### Animal Ethics
The following information was supplied relating to ethical approvals (i.e., approving body and any reference numbers):

Harvard University Institutional Animal Care and Use Committee provided full approval for this protocol 14-08-217.

### Data Availability
The raw data has been supplied as a Supplementary File.

### Supplemental Information
Supplemental information for this article can be found online at http://dx.doi.org/10.7717/peerj.2876#supplemental-information.

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
