# Peer review of "Handling stress may confound murine gut microbiota studies"

_PeerJ, doi:10.7717/peerj.2876_

## Round 0.1 · original submission · Major Revisions

The reviewers have expressed several concerns that need to be addressed before the work can be considered further. Please revise based on their comments. I look forward to the revised version.

Reviewer 1 ·

Basic reporting

Please check the format and content for a manuscript, includes the abstract section, method section, and referencing styles required by the Journal.

Experimental design

No comments

Validity of the findings

No comments

Additional comments

Abstract:
-Content need to be improved. Authors should provide brief introduction only, clear study’s objective, sufficient information on method used in the study, result with statistical p-value(s) and overall conclusion. Abstract should be in one paragraph.

Introduction:
-Please check the format of references’ citation, eg., line 41, Walter and Ley should it be Walter & Ley? Oh et al. 2010, should it be spell out if it is first time appear in text? Please check for other citations and make sure there are consistent with the requirement of the Journal.

Statistical analysis:
-Since R-studio is used in the analysis, I suggest author to perform ANOVA using R-studio rather than using Excel. In Excel, I believe it does not provide pos hoc test. It provides the global F-test only.
-Since the sample size is small <30 per group, normality of the data is in doubt, I suggest authors to use kruskal wallis test rather than ANOVA.
-line 226, wrong statement “sample sizes were quite skewed..” should be “data for variable ‘A’, ‘B’… are skewed”
-if non-parametric test such as Wilcoxon tests are used, it is unnecessary to conduct Monte Carlo simulations. Authors should justify the reason why simulations are needed in the statistical analysis.

Results:
-line 275, …significant differences in the means….. should be in the Median. Wilcoxon rank sum test is a non-parametric test, median (IQR) should be presented along with the text result.
-line 607, in table 1 and 2, median (IQR) should be reported instead of mean.

Reviewer 2 ·

Basic reporting

This is an interesting study. Hypothesis is missing in the manuscript.

Experimental design

The in vivo study has been planned in a very careful way. Confounding factors have been included in the experimental design.

Validity of the findings

- Due to the very small sample size for both treatment vs controlled groups (7 vs 22), the validity of the findings needs to be handled very carefully.

- Also, please consider to log-transform your data for statistical analysis to raise the normality. However, alternatively, the authors used Wilcoxon test to minimize the errors.

Additional comments

Please detail future prospects for the study, and its generalization for studying the relationship between human breastmilk composition and neuro-behavioral development is somewhat lacking.

Reviewer 3 ·

Basic reporting

No Comments

Experimental design

1. Blood corticosterone or cortisol level should be examined because they are potential indicater of stress.

Validity of the findings

1.The authors show significant difference between non-handled and handled pups (Figure 3 and Figure 6). However, non-handled group contains active or passive administration group and also FOS and water groups. These groups should be treated separately, therefore please confirm the statistical analysis.

2. I am confused about the data presented in Fig 3 - 6 and Table 1, as in its present form these data lack units. What “Exponential cell count” means? Also, It is mysterious that some samples contains more Bacteroides than Bacteroidetes although Bacteroides is a member of Bacteroidetes.

Additional comments

In the manuscript “Handling stress may confound murine gut microbiota studies”, Allen-Blevins et al. focus on a less invasive technique for administration to early-life mice and examine the influence of the technique on gut-microbiota. They show that both active and passive method of solution administration could change the microbiota, indicating the difficulty to handle for dosing without stress. The work is potentially important and will be of interest to readers of the journal. However, the conclusion in this article is poorly supported by these data. In addition to comments in the 3 areas above, the following points should be clarified.

1. Line 76-80: The efficacy of probiotics should be introduced at strain level, not species level. Not all strains belong to genus Bifidobcterium can produce GABA.

2. Line 110: it is ambiguous why the authors used FOS because the main target seems to be HMO in this study.

3. Line 269: “….from pups (Figure 2)”; Please confirm Whether Figure 2 is correct.

4. Although the author stated “Synbiotics, a mixture of substrate….. stress bacteria (line 390~391)”, where is the data about synbiotics? There is a little discussion about synbiotics (line350~364), but it is difficult to lead to the conclusion described in this article.

5. The authors investigated a part of gut microbiota, especially focused on Bifidobacterium and Bacteroides because of their potential roles in neurodevelopment. However, it is still unclear the relationship between gut microbiota composition and neurodevelopment. Therefore, the composition of a whole gut microbiota should be investigated.

6. Overall, the manuscript could use some help with language structure. The figure legends are well written, however the main text requires some editorial assistance for sentence structure and clarity.

---

## Round 0.2 · accepted · Accept

We are satisfied with the current revision.

Reviewer 2 ·

Basic reporting

The authors have addressed all comments/suggestions. I am satistifed with the current revised manuscript.

Experimental design

As above.

Validity of the findings

As above.

Additional comments

The current version is acceptable and merit for publication.